# Mitochondrial Dysfunction Affects the Synovium of Patients with Rheumatoid Arthritis and Osteoarthritis Differently

**DOI:** 10.3390/ijms23147553

**Published:** 2022-07-07

**Authors:** Péter Jávor, Attila Mácsai, Edina Butt, Bálint Baráth, Dávid Kurszán Jász, Tamara Horváth, Bence Baráth, Ákos Csonka, László Török, Endre Varga, Petra Hartmann

**Affiliations:** 1Department of Traumatology, University of Szeged, 6720 Szeged, Hungary; javorpeter25@gmail.com (P.J.); macsai.attila@med.u-szeged.hu (A.M.); butt.edina@gmail.com (E.B.); bbalint123@gmail.com (B.B.); csonka.akos@med.u-szeged.hu (Á.C.); torok.laszlo@med.u-szeged.hu (L.T.); varga.endre@med.u-szeged.hu (E.V.); 2Institute of Surgical Research, University of Szeged, 6720 Szeged, Hungary; jaszdkurszan@gmail.com (D.K.J.); htamara88@gmail.com (T.H.); 3Department of Pathology, University of Szeged, 6720 Szeged, Hungary; barath.bence@med.u-szeged.hu; 4Department of Sports Medicine, University of Szeged, 6720 Szeged, Hungary

**Keywords:** rheumatoid arthritis, osteoarthritis, mitochondrial respiration, complex I, cytochrome C

## Abstract

There is growing evidence regarding the role of mitochondrial dysfunction in osteoarthritis (OA) and rheumatoid arthritis (RA). However, quantitative comparison of synovial mitochondrial derangements in these main arthritis forms is missing. A prospective clinical study was conducted on adult patients undergoing knee surgery. Patients were allocated into RA and OA groups based on disease-specific clinical scores, while patients without arthritis served as controls. Synovial samples were subjected to high-resolution respirometry to analyze mitochondrial functions. From the total of 814 patients, 109 cases were enrolled into the study (24 RA, 47 OA, and 38 control patients) between 1 September 2019 and 31 December 2021. The decrease in complex I-linked respiration and dyscoupling of mitochondria were characteristics of RA patients, while both arthritis groups displayed reduced OxPhos activity compared to the control group. However, no significant difference was found in complex II-related activity between the OA and RA groups. The cytochrome C release and H_2_O_2_ formation were increased in both arthritis groups. Mitochondrial dysfunction was present in both arthritis groups; however, to a different extent. Consequently, mitochondrial protective agents may have major benefits for arthritis patients. Based on our current study, we recommend focusing on respiratory complex I in rheumatoid arthritis research.

## 1. Introduction

Arthritis is a collective term for inflammatory diseases affecting the joints with distinct etiologies, ranging from the most common form osteoarthritis (OA) to rheumatoid arthritis (RA). Joint dysfunction and pain are common characteristics of all arthritis forms [1]. A growing body of evidence implicates the role of mitochondrial damage and dysfunction in OA and RA [2,3,4,5,6]. Mitochondria are both targets and sources of inflammation-associated injury in the synovial membrane of patients, hence injury and necrosis of synoviocytes induce the release of mitochondrial damage-associated molecular patterns (DAMPs) from damaged mitochondria [7]. DAMPs trigger innate and adaptive immune responses; they promote the release of proinflammatory mediators and the aggregation of inflammatory cells. Mitochondrial dysfunction has also been implicated in the pathogenesis of primary and post-traumatic OA [8]. Some haplogroups of mitochondrial respiratory genes, and as a consequence, mitochondrial respiratory activity, are closely associated with higher prevalence of OA and RA, resulting in an increased genetic predisposition to the development of arthritis [9].

RA and OA are common forms of arthritis sharing several similarities despite their different pathogenesis. Chronic synovitis, dysregulation of synovial functions, and progressive destruction of articular cartilage occur in both disorders. In contrast to OA, RA displays more prominent inflammatory processes, and it is also characterized by the presence of an invasive synovial membrane (pannus tissue) [10]. With regards to the etiology, OA is induced by mechanical injuries, while RA is considered a multifactorial, multigenic autoimmune disease [2]. Although the contribution of mitochondrial impairment is well known in both diseases, quantitative clinical data on its extent is rare in the literature.

As mitochondria are also involved in the development of inflammation and inflammatory damage, they can be considered as future therapeutic targets for OA and RA [11,12]. However, in spite of intensive research efforts, the etiology of arthritis-related mitochondrial dysfunction is not completely understood [13], and novel disease-modifying regimens are still not available. Furthermore, the characteristics of mitochondrial derangements in OA and RA have not yet been compared. For these reasons, mitochondrial signaling pathways in the pathophysiology of joint diseases need further investigation, mainly human studies [5,13,14].

According to previous studies, mitochondrial dysfunction may affect several pathways that have been implicated in joint degradation, including hypoxia-induced signaling pathways of synovial epithelial cells, defective chondrocyte biosynthesis, and growth responses [3,4,15]. Impaired electron transport through mitochondrial complexes can also be accompanied by higher reactive oxygen species (ROS) production and lower ATP levels in chondrocytes of OA patients, as compared with healthy patients [16]. Studies demonstrate that the synovium may show significant changes prior to subchondral and cartilage degeneration, suggesting its initiative role in joint failure [17]. The arthrogenic role of the synovium has been confirmed by demonstrating that the transfer of isolated human RA synovial fibroblasts into knee joints of severe combined immunodeficiency (SCID) mice induced chronic arthritis [18]. The deteriorative pathways are suspected to be promoted by the derangements of the mitochondrial respiratory chain complexes (C I-IV), altered ATP-synthesis, cytochrome C release, and increased oxidative and nitrosative stress in the synovium [4,19].

Our present clinical study investigated the role of synovial mitochondrial dysfunction in RA and OA. We provided quantitative clinical data making the two arthritis types comparable with respect to mitochondrial dysfunction. Mitochondrial respiratory activities, cytochrome C release, and pathways of ROS production were studied in homogenates of synovial epithelial cells of RA, OA, and healthy control patients, respectively. In order to give a complex analysis of the joint status of the participants, histopathological examinations and measurements of proinflammatory cytokine levels were also performed. We hope that our results can add to the relatively small amount of scientific information available in this new, promising field of arthritis research.

## 2. Results

### 2.1. Patient Characteristics

A total of 814 patients underwent knee joint surgery between 1 September 2019 and 31 December 2021 at the Traumatology Department of the University of Szeged. Pediatric patients (*n* = 97), patients with infective/septic conditions (*n* = 58), multiple injuries (*n* = 96), compliance problems (*n* = 29), incomplete datasets (e.g., an adequate anamnesis could not be obtained) (*n* = 24), medication with mitochondrial toxicity (eg., valproic acid), and/or any potential distorting effect on our results (*n* = 80) were excluded. In 242 cases, surgery was performed within six weeks post-injury, which entailed exclusion. Seventy-nine people refused to participate. Consequently, inclusion criteria were met in 109 cases, in which 24 patients suffered from RA and 47 from OA. The control group consisted of 38 patients, without a history of OA or RA, and with a need for surgery due to trauma. The mean age of all participants was 47 ± 21 years, 52 ± 11 years in the OA, 49 ± 15, and 45 ± 8 in the RA and control group. The gender ratio was balanced. The members of the OA group had a higher mean body mass index (BMI) compared to the control group. Furthermore, more than 50% of the OA patients had a BMI ≥ 30. As a well-known fact, a BMI ≥ 30 doubles the risk of knee OA compared to normal weight or underweight individuals [20]. Most patients in the OA group suffered from moderate to severe (Kellgren–Lawrence (KL) grade 3–4) joint degeneration. Despite the advanced stages of the disease, only 38% of OA patients reported a habitual, daily intake of NSAIDs. Prior to surgery, inflammatory markers (c-reactive protein (CRP) and white blood cell count (WBC)) did not indicate an acute flare up of OA; however, a slightly elevated CRP (>5 mg/L, <25 mg/L) was measured in more than 50% of OA patients. The most common comorbidity of the study population was primary hypertension, affecting 24% of all participants. The characteristics of the patients are shown in Table 1.

### 2.2. Mitochondrial Functional Measurements

Changes in mitochondrial respiratory functions were evaluated in the presence of glutamate and malate or succinate, in order to differentiate between C I- and C II-based activity. Arthritis groups displayed significantly reduced C I activity (29.7 ± 5.7 pmol/s/mL in OA and 12.8 ± 5.4 pmol/s/mL in RA) and reduced OxPhos (65.9 ± 7 in OA and 37.3 ± 8.2 pmol/s/mL in RA) in the presence of a saturating amount of ADP compared to the control group (39.2 ± 6.8 pmol/s/mL and 86 ± 15.5 pmol/s/mL, respectively). The decrease in C I activity and the related OxPhos was significantly higher in the synovial samples of RA patients than in OA patients. However, in the presence of the C II substrate succinate, synovial mitochondria displayed similar respiratory activity in all study groups (115.2 ± 26.2 pmol/s/mL in control, 106.2 ± 19.8 pmol/s/mL in OA, and 83.4 ± 17.0 pmol/s/mL in RA). Coupled respiration was determined by the drop in respiration after oligomycin-induced inhibition of C V-associated ATP synthase. The respiratory acceptor control ratio (RCR), expressed as the OxPhos/LEAK ratio, was calculated to quantify changes in the coupling of the ETC [21]. In comparison with the control group, the coupled respiration and the RCR both indicated a significant impairment of the ETC in the RA group (13.4 ± 9.9 pmol/s/mL with 1.8 ± 0.7 RCR). However, no significant difference was found in coupling of mitochondria between the OA and the control group (78.8 ± 12.9 pmol/s/mL and 99.2 ± 18.9 pmol/s/mL) or RCR (4.9 ± 1.1 and 4.4 ± 0.5 RCR, respectively) (Figure 1).

### 2.3. Cytochrome C Release and ETC

The mitochondrial cytochrome C release and ETC are both indicators of mitochondrial membrane damage. Cytochrome C release was assessed by adding exogenous cytochrome C to the sample in the presence of glutamate and malate or succinate. The maximal respiratory capacity of ETC was induced by the protonophore FCCP. The OA group exhibited a significantly higher response in oxygen consumption compared to the control group, indicating lower initial levels of cytochrome C. Furthermore, a significant increase could be detected in the RA group compared with both the control and OA groups in cytochrome C release. The ETC was significantly lower in the RA group compared to the OA and control groups (Figure 2).

### 2.4. Mitochondrial Hydrogen Peroxide (H_2_O_2_) Production and Biochemical Analyses

There was a significant rise in the mitochondrial H_2_O_2_ level as a marker of reactive oxygen species (ROS) (i.e., superoxide anion) production in the arthritis groups (2.2 ± 0.7 pmol/s/mg protein in OA and 3.4 ± 0.8 pmol/s/mg protein in RA) compared to the control group (1.5 ± 0.3 pmol/s/mg). The free radical leak also increased in the arthritis groups; however, this rise in the RA group was significantly higher than in the OA group (Figure 3A). Xanthine oxidoreductase (XOR) and myeloperoxidase (MPO) activities were measured from synovial tissue, while nitrotyrosine (NT) levels were determined from synovial fluid as indicators of tissue damage. Significantly higher XOR activities were measured in the synovial membrane homogenates of RA and OA patients compared to the control group. Significant difference in XOR activity also occurred between the RA and OA groups. The synovial tissues of RA patients displayed significantly higher MPO activity compared to the synovial tissues of OA patients and healthy individuals. In the RA group, significant elevation of NT was present, relative to both the OA and control groups (Figure 3B).

### 2.5. Histopathological Evaluation

A confocal laser scanning endomicroscope (CLSEM) and haematoxylin and eosin (H&E) staining were used to validate the proper assignment of participants into study groups. Histological assessment was performed independently and blindly on coded slides by two investigators (P.J. and P.H.) using a previously described 0–4-grade histological scoring system, representing a composite of the extent of angiogenesis and fibrosis. Additionally, a thickened synovial membrane, increased cellularity, and mild lymphocytic infiltration occurred in the H&E-stained sections of samples from patients suffering from OA (Figure 4F). More prominent lymphocytic infiltration, fibrosis, and in some cases even extensive fibrosis, could be observed in RA samples (Figure 4F).

### 2.6. Proinflammatory Cytokines in the Synovial Fluid

TNF-α and RANKL levels in the synovial fluid samples of the participants were measured using chemiluminescence assays. The concentration of RANKL was significantly increased in the RA group compared to the control group, while TNF-α levels in the RA group were significantly higher compared to not only the control group, but also the OA group (Figure 4H,I).

## 3. Discussion

Our present study provides quantitative clinical data on mitochondrial derangements in RA and OA. Systemic disease activity was evaluated by measuring serum CRP concentrations, proinflammatory cytokines in the synovial fluid samples, and clinical scoring (KL classification, 2010 American College of Rheumatology/European League Against Rheumatism (ACR/EULAR) criteria) of our patients. Additionally, histopathological evaluations and in vivo histology with CLSEM were also performed on tissue samples to detect the histological characteristics of RA and OA. Our results complied with the literature, as RA patients had significantly higher levels of proinflammatory cytokines and hyperactive pathways of ROS production. Additionally, histopathology revealed signs of chronic inflammation such as neoangiogenesis, increased mean lining thickness, and fibrosis in both OA and RA patients; however, to a much higher extent in the latter [22].

The capacities of the ETC complexes in the study groups were tested with high-resolution respirometry. Interestingly, C I activity was strongly diminished in the RA group, while there was no significant difference in C II activity between the groups. This result highlights the central role of C I in mitochondrial dysfunction in RA, setting it as the main target for future RA therapies. C I is the first, largest, and most complicated component of the respiratory chain. Furthermore, as the major entry point for electrons to the respiratory chain, C I is considered as the rate-limiting factor in overall respiration [23]. Additionally, it generates significant levels of ROS, especially during reverse electron transport [21,24]. As a consequence, C I has already stood in the focus of researchers investigating potential mitochondrial protective drugs. According to this, a recent study demonstrated the C I specific ROS-production inhibiting effect of OP2113 (5-(4-Methoxyphenyl)-3H-1,2-dithiole-3-thione, CAS 532-11-6), and highlighted its role in therapies for Parkinson’s and Alzheimer’s diseases [25]. Additional to aging-related diseases, based on our findings, we emphasize the importance of C I related ROS blockers in autoimmune diseases, including RA.

Compared to patients suffering from RA, members of the OA group displayed a milder, but still notable decrease in C I activity compared to healthy individuals in the control group. Again, a significant deficit in C II activity did not occur. This result complies with the literature accentuating the importance of C I in aging-related diseases. Based on our findings, we suggest putting the focus on C I-specific pharmaceutical agents in research for novel therapeutic options for OA. Beyond highly effective conservative treatment options, prognostic biomarkers for disease progression and early-stage osteoarthritis are lacking [26]. A recent study suggested that certain mitochondrial DNA (mtDNA) haplogroups may be suitable to aid the recognition of early-stage OA, and they may also have a prognostic potential for disease progression [11]. Cytochrome C serves as an electron shuttle in the respiratory chain in the mitochondrial intermembrane space. It is well known that inflammatory stimuli promote the release of cytochrome C from the intermembrane space to the cytosol, where it facilitates apoptosis through initiating the proteolytic maturation of caspases [27,28]. Based on our findings, further studies investigating other mitochondrial parameters, such as C I activity or cytochrome C release, as prognostic markers for disease progression are warranted.

Bone destruction is a common feature of inflammatory arthritis and is mediated by osteoclasts. Both synovium secreted soluble factors, namely RANKL and TNF-α, have the ability to directly induce osteoclastogenesis. RANKL and TNF-α levels were significantly elevated in RA patients in our study [29]. Mitochondria are considered an early target in TNFα-induced cytotoxicity, since TNF-α induces mitochondrial damage through suppression of respiratory C I–IV [30]. Aberrant expression of RANKL, an inducer of osteoclast differentiation, has been linked to synovial fibroblasts and bone pathology in RA. Moreover, increased ROS production results in oxidation of mitochondrial lipids, sulfhydryl groups, and iron sulfur complexes of mitochondrial respiratory enzymes, leading to the impairment of OxPhos [31].

Our present study demonstrated increased synovial tissue levels of XOR and NT in both arthritis study groups compared to healthy controls, while MPO was elevated only in RA patients. XOR is an important enzyme in purine catabolism. Besides converting xanthine to uric acid, XOR also catalyzes the reduction in nitrates and converts nitrites into nitric oxide. This process is accompanied by the production of ROS, which can result in cellular disruption. Nitric oxide reacts in a diffusion-limited manner with superoxide anion to form peroxynitrite, a powerful oxidant that has been linked to tissue injury. As a stable end product of peroxynitrite formation, NT reflects disease-associated tissue damage in the joint, thereby serving as a potential biomarker for disease activity [32,33]. MPO plays a central role in the pathogenesis of autoimmune inflammation through initiating the production of hypochlorous acid (HClO) and other reactive agents, causing tissue damage. According to the literature, elevated MPO levels are observed in a number of autoimmune diseases, including multiple sclerosis and RA [34].

## 4. Materials and Methods

### 4.1. Ethical Approval

The study was conducted in accordance with the Declaration of Helsinki and has been approved by the local medical ethics committee at the University of Szeged under reference number 85/2019-SZTE.

### 4.2. Study Design

The prospective clinical study was conducted at a single, level I trauma center (Department of Traumatology at the University of Szeged) between 1 September 2019 and 31 December 2021.

Data were collected from consecutive, adult patients (age > 18 years) with signed informed consent, undergoing open or arthroscopic knee joint surgery. Pediatric patients and patients with multiple injuries, septic conditions, inadequate compliance, or an incomplete dataset were excluded.

Labor parameters (WBC, CRP, total protein (TP)), clinically relevant information (age, gender, height, weight, body mass index (BMI), International Statistical Classification of Diseases and Related Health Problems (ICD) codes, comorbidities, previous joint injuries, medication, etiology, current activity of OA, and type of operation) were extracted from an electronic database (MedSolution). Additional information regarding subjective life quality, the level of everyday joint pain (according to VAS), and the use of orthoses and walking aids were obtained from a detailed questionnaire.

### 4.3. Patient Allocation

Participants were allocated into RA, OA, and control groups based on their medical documentation, ACR/EULAR rheumatoid arthritis classification criteria, and KL classification. Patients were allocated into the RA group if they had already been diagnosed with RA and had fulfilled the ACR/EULAR score ≥ 6 criterion. For the diagnosis of OA, the KL classification is the most widely used radiographic clinical tool [35,36]. AP knee radiographs were graded from 0 to 4, according to the KL-criteria, by 2 independent orthopedic trauma experts (Á.C., A.M.). A grade > 1 entailed an allocation into the OA group, independent from the etiology of the disease (both primary and post-traumatic cases were included). The control group consisted of patients who did not have RA in their patient history and had a KL grade ≤ 1.

### 4.4. Sampling

Synovial fluid and tissue samples from the knee joint synovium were taken with the permission and signed consent of the patients. Synovial fluid was aspirated prior to incision with an 18 G needle and placed into Eppendorf tubes. Synovium tissue samples of 1 × 1 cm in size were obtained from the suprapatellar pouch, sparing the Hoffa’s fat pad. The samples were placed into 4% (*v*/*v*) neutral buffered formalin and phosphate buffered saline (PBS) and were transported on ice directly to undergo biochemical measurements, histopathological evaluation, and high-resolution respirometry, respectively [37].

### 4.5. Examination of Mitochondrial Functions

The efficacy of mitochondrial respiration was assessed with high-resolution respirometry (Oxygraph-2k, Oroboros Instruments, Innsbruck, Austria) in synovium homogenates. The tissue samples were homogenized in 200 µL of MiR05 respiration medium [38], with a glass Potter–Elvehjem homogenizer. Subsequently, the homogenates were weighed into the detection chambers, 50 µL in each, which were calibrated to 200 nmol/mL oxygen concentration in room air. Our protocol was used to explore the relative contribution of respiratory complexes to the electron transport chain (ETC) and the oxidative phosphorylation capacity (OxPhos) of mitochondria. First, the steady-state basal oxygen consumption of the homogenates (respiratory flux) was measured. Glutamate (10 mM) and malate (2 mM) were used in combination to induce C I-linked respiration. The oxidative phosphorylation capacity was measured by adding saturating ADP (5 mM final concentration). Then, C II-linked respiration (10 mM succinate-fuelled) was determined. Cytochrome C release (an indicator of inner mitochondrial membrane damage) was determined as described previously [39]. The intactness of the inner mitochondrial membrane was assessed after adding cytochrome C (10 μM). Leak respiration (LEAK) was measured in the presence of C V inhibitor oligomycin (Omy) (1 mM). Thereafter, protonophore agent carbonyl cyanide p-trifluoromethoxy-phenyl-hydrazine (FCCP) (0.5 μM) was added to measure the maximal ETC capacity. Finally, residual oxygen consumption (ROX) was determined by adding 1 μM rotenone (Rot) and 1 μM antimycin-A (Ama). The mitochondrial H_2_O_2_ release was monitored fluorimetrically with the Amplex Red/horseradish peroxidase system, whereby Amplex Red (non-fluorescent) is oxidized to resorufin. Oxidizing substrates (20 mmol/L glutamate, 10 mmol/L malate, 10 mmol/L succinate, and 5 mmol/L ADP) were added to the mitochondria. The reverse electron transport (RET)-initiated H_2_O_2_ flux was measured when mitochondria were blocked by the addition of 1 µmol/L rotenone. The residual oxygen consumption was estimated after addition of 1 μmol/L antimycin-A to exclude the effects of oxidative side reactions. H_2_O_2_ production was calibrated with known amounts of H_2_O_2_. Then, free radical leak was also determined as the percentage of oxygen consumption diverted to the production of H_2_O_2_ in State 3. All reagents of respirometry, including respiratory substrates and inhibitors, were purchased from Sigma Aldrich (St. Louis, MO, USA). Manual titration of these substances for 2 mL volume was carried out with Hamilton syringes. (Details on exact volumes and concentrations can be found at https://wiki.oroboros.at/images/f/fc/Gnaiger_2014_Mitochondr_Physiol_Network_MitoPathways.pdf, accessed on 13 June 2022).

### 4.6. Biochemical Analyses

#### 4.6.1. Tissue Xanthine Oxidoreductase (XOR) Activity

Synovial membrane samples were homogenized in a phosphate buffer (pH 7.4) containing 50 mM Tris-HCl, 0.1 mM EDTA, 0.5 mM dithiotreitol, 1 mM phenylmethylsulfonyl fluoride, 10 μg mL^−1^ soybean trypsin inhibitor, and 10 μg ml^−1^ leupeptin. The homogenate was centrifuged at 4 °C for 20 min at 24,000× *g*, and the supernatant was loaded into centrifugal concentrator tubes. The activity of XOR was determined in the ultrafiltered supernatant by a fluorometric kinetic assay based on the conversion of pterine to isoxanthopterine in the presence (total XOR) or absence (XO activity) of the electron acceptor methylene blue (Sigma-Aldrich, Budapest, Hungary) [40].

#### 4.6.2. Tissue Myeloperoxidase (MPO) Activity

The MPO activity was measured in synovial tissue according to the method of Kuebler et al. [41]. Briefly, the synovial tissue was homogenized with a Tris-HCl buffer (0.1 M, pH 7.4) containing 0.1 M polymethylsulfonyl fluoride to block tissue proteases, and then centrifuged at 4 °C for 20 min. at 24,000× *g*. The MPO activity of the samples was measured at 450 nm (UV-1601 spectrophotometer; Shimadzu, Japan). The data referred to the protein content.

#### 4.6.3. Nitrotyrosine (NT) Levels

Free nitrotyrosine, a marker of peroxynitrite generation, was measured with an enzyme-linked immunosorbent assay (Cayman Chemical, Ann Arbor, MI, USA). The synovial fluid was centrifuged at 15,000× *g*. The supernatants were collected and incubated overnight with antinitrotyrosine rabbit IgG and nitrotyrosine acetylcholinesterase tracer (Cayman Chemical, Ann Arbor, MI, USA, Cat. No: 414006) in precoated (mouse antirabbit IgG, Cayman Chemical, Ann Arbor, MI, USA, Cat. No: 40004) microplates, followed by development with Ellman’s reagent (Cayman Chemical, Ann Arbor, MI, USA, Cat. No: 400050). Nitrotyrosine content was normalized to the protein content of the homogenate and expressed in ng/mg.

#### 4.6.4. Laboratory Testing of Synovial Fluid

Proinflammatory cytokines tumor necrosis factor alpha (TNF-α; Sigma-Aldrich, Budapest, Hungary, Cat. No: SCP0254) and receptor activator of nuclear factor kappa-beta ligand (RANKL; Sigma-Aldrich, Budapest, Hungary, Cat. No: HRNKLMAG-51K-01) levels were measured in the synovial fluid samples using a commercially available ELISA kit (Sigma-Aldrich, Budapest, Hungary, Cat. No: RAB1073) according to the manufacturer’s instructions.

### 4.7. Histopathological Analysis

Intraoperatively harvested synovium biopsies were assessed with light microscopy and confocal imaging. Confocal imaging with a laser scanning endomicroscope (CLSEM, FIVE1 system, Optiscan, Victoria, Australia) was started immediately after retrieving the synovial sample. The rigid confocal probe (excitation wavelength 488 nm; emission detected at 505–585 nm) was mounted on a specially designed metal frame and gently pressed onto the inner surface of the joint capsule (1 scan/image, 1024 × 512 pixels and 475 × 475 μm per image). For the in vivo staining, 0.01% acriflavine (Sigma-Aldrich, Budapest, Hungary) was applied topically [39]. The analysis was performed twice, separately by the same investigator (PH), using a semiquantitative histology score (S0–S4) based on widening of synovial lining, neoangiogenesis, collagen fibre disorganization, and fragmentation, as described previously [42]. For traditional light microscopy, the samples were fixed in a buffered paraformaldehyde solution (4%) and embedded in paraffin. 5-μm thick sections were cut and then stained with H&E.

### 4.8. Statistical Analysis

The statistical analysis was performed with SigmaStat 13.0 statistical software (Jandel Corporation, San Rafael, CA, USA). Normal distribution was tested with the Shapiro-Wilk test. In case of a normal distribution, one-way ANOVA with Tukey’s test was used and data were expressed as mean ±  SD. *p*  <  0.05 was considered as statistically significant.

## 5. Conclusions

Our present study provides quantitative clinical data making RA and OA comparable with respect to mitochondrial dysfunction. The characteristics of mitochondrial derangements in OA and RA share several similarities, including increased ROS production and C I related diminution of respiratory activity. Differences are reflected mainly in the extent of respiratory deficit (RA > OA), ROS formation (RA > OA), and cytochrome C release (RA > OA). The mitochondrial ETC damage and dyscoupled electron transport were more characteristic of RA. According to our findings, in contrast to C I, C II does not play a central role in altered mitochondrial functions in the two arthritis types. Consequently, focusing on C I-specific mitochondrial protective agents in OA and RA research may have major benefits.

## Figures and Tables

**Figure 1 ijms-23-07553-f001:**
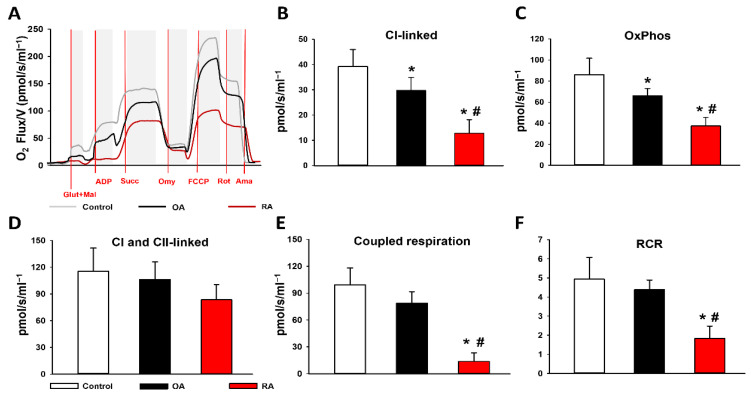
Mitochondrial functional measurements. (**A**) Representative records of mitochondrial oxygen consumption measured by HRR. (**B**) Complex I (CI) linked (NADH-generating substrates: glutamate (Glut) and malate (Mal)-dependent) respiration. (**C**) OxPhos (ADP-dependent) respiration in the presence of saturating levels of substrates. (**D**) CI and complex II (CII)-linked (added succinate (Succ) respiration. (**E**) Coupled respiration demonstrates the difference before and after oligomycin (Omy). (**F**) Respiratory control ratio (RCR) expressed as the ratio of OxPhos/LEAK respiration. The RA group is marked with red. Black columns represent the OA group. The control group is marked with white. Data are presented as means ± SD. * *p* < 0.05 vs. control; # *p* < 0.05 vs. OA (one-way ANOVA, Tukey’s test).

**Figure 2 ijms-23-07553-f002:**
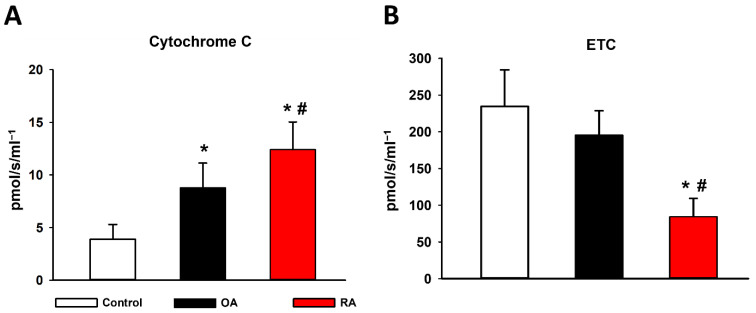
Cytochrome C induced respiration and the ETC of synovial mitochondria. (**A**) Cytochrome C release assessed by adding exogenous cytochrome C to the sample in the presence of glutamate (Glut) and malate (Mal) or succinate (Succ). (**B**) Electron transport chain (ETC) induced by the protonophore p-trifluoromethoxy-phenyl-hydrazine (FCCP). The RA group is marked with red. Black represents the OA group. The control group is marked with white. Data are presented as means ± SD. * *p* < 0.05 vs. control; # *p* < 0.05 vs. OA (one-way ANOVA, Tukey’s test).

**Figure 3 ijms-23-07553-f003:**
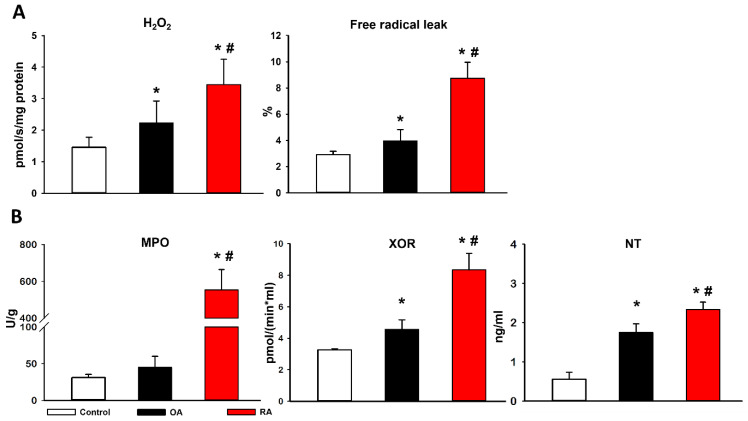
(**A**) Mitochondrial reactive oxygen species (ROS) production. The left chart demonstrates the mitochondrial H_2_O_2_ release as a marker of reactive oxygen species (ROS) (i.e., superoxide anion) production, monitored fluorimetrically with the Amplex Red/horseradish peroxidase system. The right chart shows the free radical leak as the percentage of oxygen consumption diverted to the production of H_2_O_2_ in OxPhos state. (**B**) Biochemical analyses of synovial tissue. The charts demonstrate myeloperoxidase (MPO), xanthine oxidoreductase (XOR), and nitrotyrosine (NT) enzyme activities. Black represents the OA group. The control group is marked with white. Data are presented as means ± SD. * *p* < 0.05 vs. control; # *p* < 0.05 vs. OA (one-way ANOVA, Tukey’s test).

**Figure 4 ijms-23-07553-f004:**
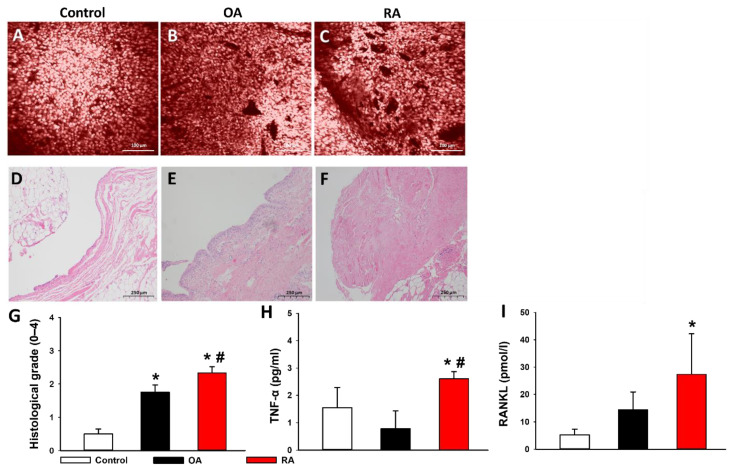
Histological changes and inflammatory cytokines in the synovium. (**A**–**C**) Tissue sections show the results of in vivo confocal laser scanning endomicroscopy with acriflavine labeling. The bar represents 100 μm. (**D**–**F**) Histological sections stained with hematoxylin and eosin. (**A**) Healthy synovium with rounded synoviocytes. No signs of inflammation-related angiogenesis and fibrosis can be seen. (**B**) Osteoarthritic synovium with moderate angiogenesis due to low-grade chronic inflammation. (**C**) Synovium of a patient suffering from RA. The large number of vascular cross-sections refers to inflammation-related angiogenesis. Scarring occurs as a result of chronic synovitis. (**D**) Joint capsule section with synovial membrane from a healthy joint. Flattened mast cells constitute a single cell layer. The lamina propria is poor in cells and rich in connective tissue fibers. Adipocytes and cross-sections of capillaries can be observed in the deeper layers. (**E**) Joint capsule sample from osteoarthritic joint. The synovial membrane is thickened, consists of 3–4 cell layers. Increased cellularity occurs, partly due to the mild lymphocytic infiltration that can also be observed in the lamina propria. (**F**) Joint capsule sample of a patient suffering from RA. As a result of extensive scarring, the synovial membrane is unrecognizable. Fibrosis affects more than 50% of the stroma. (**G**) Histological grading is performed by allocating a composite number to the groups based on the extent of angiogenesis and fibrosis (0–4 grade). (**H**) TNF-α levels in the synovial fluid. (**I**) RANKL levels in the synovial fluid. RA group is marked with red. Black represents the OA group. The control group is marked with white. Data are presented as means ± SD. * *p* < 0.05 vs. control; # *p* < 0.05 vs. OA (one-way ANOVA, Tukey’s test). TNF-α = Tumor Necrosis Factor-α, RANKL = Receptor Activator of Nuclear factor-κB Ligand.

**Table 1 ijms-23-07553-t001:** Patient characteristics. The patients were graded according to the Kellgren–Lawrence classification by two independent orthopedic trauma surgeons. VAS refers to the chronic pain in the joint affected by OA. Acute pain due to acute trauma was not considered. Daily use of NSAIDs refers to habitual drug intake. Taking NSAIDs for a short period after trauma was not regarded as daily use. Crutches, rollators, and wheelchairs were categorized as walking aids. Labor parameters were measured prior to joint surgery. CRP levels were not measurable under 2 mg/L, thus a CRP < 2 mg/L was considered as 0. OA = osteoarthritis; SD = standard deviation; BMI = body mass index; IQR = interquartile range; VAS = visual analog scale; ACR/EULAR = American College of Rheumatology/European League Against Rheumatism; NSAID = non-steroidal anti-inflammatory drug; WBC = white blood cell; CRP = C-reactive protein; TP = total protein; NIDDM = non-insulin dependent diabetes mellitus; and IDDM = insulin dependent diabetes mellitus.

Demographics	All Patients (*n* = 109)	RA Group (*n* = 24)	OA Group (*n* = 47)	Control Group (*n* = 38)
Age (y) (mean ± SD)	47 ± 21	49 ± 15	52 ± 11	45 ± 8
Female *n* (%)	56 (51)	19 (79)	21 (45)	16 (42)
Male *n* (%)	53 (49)	5 (21)	26 (55)	2215 (58)
**OA Risk Factors**				
Age >50 years *n* (%)	48 (44)	11 (46)	30 (64)	7 (18)
BMI (mean ± SD)	29 ± 5	28 ± 4	33 ± 6	26 ± 3
BMI ≥30 *n* (%)	32 (29)	2 (1)	25 (53)	5 (13)
Joint trauma in the anamnesis *n* (%)	64 (59)	3 (1)	23 (49)	38 (100)
**Disease Severity**				
ACR/EULAR score (mean ± SD)		7 ± 1		
ACR/EULAR score (median [IQR])		7 [6,7]		
Kellgren–Lawrence Score (mean ± SD)		4 ± 1	4 ± 1	
Kellgren–Lawrence Score (median [IQR])		4 [3,4]	4 [3,4]	
VAS (mean ± SD)		8 ± 2	5 ± 3	
Takes NSAIDs daily *n* (%)		11 (46)	18 (38)	
Needs walking aid *n* (%)		12 (50)	19 (40)	
**Labor Results**				
WBC (G/L) (mean ± SD)	9.8 ± 3.6	10.8 ± 3.3	10.3 ± 3.8	8.3 ± 3.0
WBC (G/L) (median [IQR])	9.2 [7.5–12.0]	10.1 [8.8–13.2]	10.0 [7.9–12.1]	7.6 [6.1–10.0]
CRP (mg/L) (mean ± SD)	7.0 ± 7.0	11.2 ± 8.9	7.3 ± 6.1	3.5 ± 4.3
CRP (mg/L) (median [IQR])	5.5 [2.9–8.8]	6.9 [5.4–15.6]	5.6 [3.5–10.4]	2.8 [0.0–5.5]
TP (g/L) (mean ± SD)	70.1 ± 2.9	69.2 ± 2.5	70.5 ± 3.0	70.2 ± 2.7
TP (g/L) (median [IQR])	69.4 [68.5–71.0]	72.2 [67.5–69.8]	69.6 [69.2–71.0]	69.5 [68.5–71.6]
RF positive *n* (%)		15 (63)		
**Comorbidities**				
Presence of comorbidities *n* (%)	63 (58)	21 (88)	38 (81)	4 (118)
Primary hypertension	26 (24)	6 (25)	18 (38)	2 (5)
Diabetes	12 (11)	2 (8)	8 (17)	2(5)
*NIDDM*	*9 (8)*	*2 (8)*	*5 (10)*	*2 (5)*
*IDDM*	*3 (3)*	*0 (0)*	*3 (6)*	*0 (0)*
Gout	3 (3)	1 (4)	2 (4)	0 (0)
Other	22 (31)	12 (71)	10 (32)	0 (0)
**Operation Type**				
Arthroscopy *n* (%)	47 (43)	2 (8)	12 (26)	33 (87)
Open surgery *n* (%)	62 (57)	22 (92)	35 (74)	5 (13)

## Data Availability

The data that support the findings of this study are available from the corresponding author upon reasonable request.

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
