# Peer review of "Mitochondrial Dysfunction Affects the Synovium of Patients with Rheumatoid Arthritis and Osteoarthritis Differently"

_ijms, 2022, doi:10.3390/ijms23147553_

Round 1

Reviewer 1 Report

Please see attached

Author Response

Dear Prof. Li,

Thank you for your letter dated June 29, 2022. The authors have revised the manuscript according to the Reviewer’s suggestion. Our response is provided below with references to the corresponding changes in the manuscript. We hope that the revised paper is now acceptable for publication in the Journal.

Reviewer #1

1.) Osteoarthritis and osteoarthrosis have been used synonymously in the paper. Although there are some sources that says both are interchangeable, the etiopathology of both the diseases is different (inflammation and degeneration). It is better to clarify why the authors are using interchangeably or differentiate both of them and use only.

Thank you for this comment, the Reviewer is right. It is not our intention to confuse the two terms, therefore osteoarthrosis has been replaced with osteoarthritis throughout the paper. 

2.) Figure 1, upper middle chart shows control group C I activity of 38 – 40 pmol/s/m-1, whereas the description says 61.9 ± 14.8 pmol/s/ml. There’s a discrepancy between these two and if the figure value is correct, then is it significantly different from the disease group. Please clarify.

Thank you for this remark. We have made an unfortunate mistake, some values of CI activity has been moved from the wrong row of descriptive statistics to the article. The right values of the control group are 39.2± 6.8 pmol/s/ml, as it can be seen on the graph. Levels of significance were displayed properly (p=0.039 vs. OA and p= <0.001 vs. RA). Page 7 para 2 line 135-137: “Arthritis groups displayed significantly reduced C I activity (29.7 ± 5.7 pmol/s/ml in OA and 12.8 ± 5.4 pmol/s/ml in RA) and reduced OxPhos (65.9 ± 7 in OA and 37.3 ± 8.2 pmol/s/ml in RA) in the presence of saturating amount of ADP as compared to the control group (39.2 ± 6.8 pmol/s/ml and 86 ± 15.5 pmol/s/ml, respectively).”

Figure 1, Please add values for the lower left chart in the description for clarity purpose.

Values were added to description on page 7 para 2 line 139-141: “However, in the presence of the C II substrate succinate, synovial mitochondria displayed similar respiratory activity in all study groups (115.2 ± 26.2 pmol/s/ml in control, 106.2 ± 19.8 pmol/s/ml in OA and 83.4 ± 17.0 pmol/s/ml in RA) .”

Figure 1, what is the RCR value for the control and OA groups.

RCR values of the control and OA groups are 4.9 ± 1.1 and 4.4 ± 0.5, respectively. Data were added to description page 7 para 2 line 145-147: “However, no significant difference was found in coupling of mitochondria  between the OA and the control group  (78.8 ± 12.9 pmol/s/ml and 99.2 ± 18.9 pmol/s/ml) or RCR (4.9 ± 1.1 and 4.4 ± 0.5 RCR, respectively)”

3.) Cytochrome C release and ETC tests have been performed which indicates mitochondrial membrane damage. Along with this, is there any data measuring mitochondrial membrane potential (MMP) which is a major indication of chondrocyte damage in OA.

We agree with this remark, MMP is one of the major indication of chondrocyte damage in OA. Unfortunately our ethical approval allowed us to take synovial fluid and tissue samples but not any cartilage or bone samples.

We have performed several measurements on synovial samples, and when the sample volume allowed us, we also performed MMP measurements using the fluorescent module of the respirometer, according to our previously established method (doi: 10.1111/jcmm.16498.). Unfortunately, the number of MMP measurements in the RA group did not allow us to perform statistical analysis. Consistent with previous results, our preliminary data showed that MMP in synovial mitochondria of OA patient is depolarized, which refers to mitochondrial membrane damage. Nevertheless, MMP measurement requires further investigations with adequate number of cases, which is determined by sample size calculation.

4.) Figure 4, histopathological images show increased lymphocytic infiltration in OA rather than RA, in contrary to the results stated. Moreover immunohistochemistry would be a better way to show lymphocytic infiltration or any specific cells.

Thank you for this remark. After examining multiple tissue samples, the results were described based on the data, which might not neccesarily be observed on the pictures submitted. The histological images shown in the article are representative of specific, characteristic pathological changes from three patients.

Identification of specific inflammatory cells using immunohistochemical reactions was not the main objective of the histopathological examination, hematoxylin and eosin staining provided sufficient data.

5.) In Figure 4I, why does the TNF-α levels are less in OA group compared to the control group?

It would also be useful if there are any images showing bone damage like µ-CT or any microscopic images since bone damage is the most common feature in both OA and RA.

Thank you for this comment. Indeed, TNF-α levels seems to be lower in the OA group than in the control group; however, there is no significant difference between them (P = 0,260).

The Reviewer is right, bone damage is characteristic both in OA and RA. We have not performed µ-CT, but we used conventional radiographs for Kellgren-Lawrence (KL) scoring, which is based on the bone and cartilage deformities of the joint. We have shown KL grades only in the OA group, because it is used for the diagnosis and for the evaluation of the grade of OA, but we have the values for all patients. Similarly to OA patients, the presence of osteophytes, narrowing of joint space, sclerosis and deformity of the bone ends were common in RA patients; therefore, most individuals were classied as KL 3 or 4. Now it is show in the Table.

Finally, we would like to thank the Editor and the Reviewers for their conscientious work and overall positive criticism of our manuscript. We hope that the revised paper is now acceptable for publication in the Journal.

Reviewer 2 Report

In the scientific work: “Mitochondrial dysfunction affects the synovium of patients with rheumatoid arthritis and osteoarthrosis  differently in the mechanical cushioning of joints and the maintenance of tissue homeostasis” the authors explain  the role of mitochondrial dysfunction in osteoarthritis (OA) and rheumatoid arthritis (RA) and show that a mitochondrial protective agents in arthritis research may be helpful. 

Overall, this manuscript results very interesting, the authors clearly explain the rational of the study and discussed the obtained results.

However, we would like to invite the authors  to clarify some minor points:

 1.       Please check the check punctuation and spaces;

2.       The authors introduce the topic with the description of OA, however the reference  [1]  appears a little old, the authors should add another more recent;

3.      The authors analyzed some biomarkers related to mitochondrial dysfunction, did you perferm other analyses (e.g qRT-PCR, western blotting) ?

4.       Figure 4: it is not clear the performed staining for each image, for example, please better specify acriflavine labeling what’s means;

Author Response

1.) Please check the check punctuation and spaces

Thank you for this comment. Formal and grammatical mistakes of the paper has now been corrected by a native speaking lector.

2.) The authors introduce the topic with the description of OA, however the reference [1]  appears a little old, the authors should add another more recent.

Thank you for this remark. Reference [1] has been replaced with a more recent one: Pap, T.; Korb-Pap, A., Cartilage damage in osteoarthritis and rheumatoid arthritis--two unequal siblings. Nat Rev Rheumatol 2015, 11, (10), 606-15.

3.) The authors analyzed some biomarkers related to mitochondrial dysfunction, did you perferm other analyses (e.g qRT-PCR, western blotting)?

Thank you for this question. Our primary aim was to assess the synovial mitochondrial functions in OA and RA patients. During respitometry, the mitochondrial electron transport, the oxydative phosphorilation and the efficacy of the ADP synthesis can be calculated on the basis of oxygen consumption of small (≥10 mg), freshly-taken tissue samples during arthroscopies or open surgeries. Although it would be promising to investigate the expression changes of mitochondrial respiratory genes and their association with the complex activities, these analyses were beyond the technical capabilities of the present clinical study.

4.) Figure 4: it is not clear the performed staining for each image, for example, please better specify acriflavine labeling what’s means

Now the staining for each images are better clearified in the Results section as well on page 10 para 1 lines 194-201: „CLSEM and H&E staining were used to validate the proper assignment of participants to study groups. Histological assessment of CLSEM images with acriflavine staining (Figure 4A-C) was performed using a previously described histological scoring system [21]. Accordingly, the extent of angiogenesis and fibrosis was increased in the arthritis groups; however, this was more pronounced in the RA group than in the OA group (Figure 4G).  Additionally, on H&E stained sections, a thickened synovial membrane, increased cellularity and mild lymphocytic infiltration occurred in samples from patients suffering from OA as compared to the control (Figure 4D-E). More prominent lymphocytic infiltration, fibrosis, and in some cases even extensive fibrosis could be observed in RA samples (Figure 4F).”

Acriflavin labeling is better specified in the Materials and Methods section on page 14 para 4 lines 386-387: „For the in vivo staining, 1 ml of the fluorescent dye 0.01% acriflavine (Sigma-Aldrich, Budapest, Hungary) was applied topically [40]. The surplus dye was washed off with 154 mM NaCl, then confocal imaging was performed 2 min after dye administration. The analysis was performed off line, independently and blinded on coded image slides…”

 Finally, we would like to thank the Editor and the Reviewers for their conscientious work and overall positive criticism of our manuscript. We hope that the revised paper is now acceptable for publication in the Journal.

Reviewer 3 Report

In this study, synovial fluids and tissue samples from healthy controls, OA and RA patients are differentiated with respect to mitochondirieal derangements. This descriptive study is interesting and may be published in IJMS after minor modifications.

- The manufacturer information on high-resolution respirometry should be removed from the abstract.

- In all graphs, the information about OA, RA and controll should be written under the bars. This increases the clarity.

- The graphs should also be labeled with letters in Figs. 1 and 2.

- Histological assessment was performed independently and blindly on coded slides by two investigators (P.J. and P.H.) using a previously described 0-4-grade histological scoring system, representing a composite of the extent of angiogenesis and fibrosis: information on hisological assessment in the Materials and Methods section not on the results. Cite source of histological scoring.

- Figs. 4A-E are not mentioned in the body text. Instead of Fig. A-g, I would write the groups in the images.

- The scale bar is missing

- In Fig 4I, the labeling has slipped into the axis. In Fig. 4H the asterisks are shifted.

- Pg 12 top: complexes of mitochondrial respiratory enzymes that lead to the impairment of OxPhos [31]. [31][31][31][31][31][31][31][31][31][31][31][31][31][31][30][29][28][27][27][26][25][24][23][25][24][23][22][21][20][19][19][18][17][16][15][15][15][14][14][14][13][12][11][11][10][9][8][7][6][5][4][3][2] please correct

- Inclusion criteria should be named. How many patients were screened/interviewed. How many included.

- Most order numbers and companies are missing.

Author Response

Reviewer #3

1.) The manufacturer information on high-resolution respirometry should be removed from the abstract.

Thank you for this comment. The manufacturer information on high-resolution respirometry have been removed from the abstract.

2.) In all graphs, the information about OA, RA and controll should be written under the bars. This increases the clarity.

Thank you for this remark. In order to increase the clarity we have supplied all figures with labelled control, OA and RA bars.

3.) The graphs should also be labeled with letters in Figs. 1 and 2.

Thank you for this comment. Now, graphs are labeled with letters in Figs. 1 and 2.

4.) Histological assessment was performed independently and blindly on coded slides by two investigators (P.J. and P.H.) using a previously described 0-4-grade histological scoring system, representing a composite of the extent of angiogenesis and fibrosis: information on hisological assessment in the Materials and Methods section not on the results. Cite source of histological scoring.

Thank you for this comment. Accordingly, information on histological assessment has been removed from the Results to the Materials and Methods section. The Results section has been corrected as follows on page 10 para 1 lines 194-201: „CLSEM and H&E staining were used to validate the proper assignment of participants to study groups. Histological assessment of CLSEM images with acriflavine staining (Figure 4A-C) was performed using a previously described histological scoring system [21]. Accordingly, the extent of angiogenesis and fibrosis was increased in the arthritis groups; however, this was more pronounced in the RA group than in the OA group (Figure 4H).  Additionally, on H&E stained sections, a thickened synovial membrane, increased cellularity and mild lymphocytic infiltration occurred in samples from patients suffering from OA as compared to the control (Figure 4E-F). More prominent lymphocytic infiltration, fibrosis, and in some cases even extensive fibrosis could be observed in RA samples (Figure 4G).”

Whereas, the Materials and Methods section were corrected as follows on page 14 para 4 lines 387-390:The analysis was performed off line, independently and blinded on coded image slides by two investigators (P. J. and P.H) using a semiquantitative histology score (S0-S4) based on widening of synovial lining, neoangiogenesis, collagen fibre disorganization and fragmentation, as described previously [21].” Reference [21]:Wu, J. P.; Walton, M.; Wang, A.; Anderson, P.; Wang, T.; Kirk, T. B.; Zheng, M. H., The development of confocal arthroscopy as optical histology for rotator cuff tendinopathy. J Microsc 2015, 259, (3), 269-75.

5.) Figs. 4A-E are not mentioned in the body text. Instead of Fig. A-g, I would write the groups in the images.

Thank you for noticing. Figure 4A-E are now mentioned in the Results section (see the answer above), additionally, groups are labelled above images.

6.) The scale bar is missing

Thank you for this comment, a scale bar is now added to the histological images.

7.) In Fig 4I, the labeling has slipped into the axis. In Fig. 4H the asterisks are shifted.

Thank you for noticing. It has been corrected.

8.)  Pg 12 top: complexes of mitochondrial respiratory enzymes that lead to the impairment of OxPhos [31]. [31][31][31][31][31][31][31][31][31][31][31][31][31][31][30][29][28][27][27][26][25][24][23][25][24][23][22][21][20][19][19][18][17][16][15][15][15][14][14][14][13][12][11][11][10][9][8][7][6][5][4][3][2] please correct.

Thank you for this remark. The reference mistake has been corrected.

9.) Inclusion criteria should be named. How many patients were screened/interviewed. How many included.

Patient inclusion criteria are more detailed in the Materials and methods section on page 12 para 5 line 301-312:

„4.3 Patient allocation and inclusion criteria

            Participants were allocated into RA, OA and control groups based on their medical documentation, 2010 American College of Rheumatology/European League Against Rheumatism (ACR/EULAR) rheumatoid arthritis classification criteria and Kellgren-Lawrence (KL) classification. Patients were allocated into the RA group if they have already been diagnosed with RA and fulfilled the 2010 ACR/EULAR score ≥6 criterion. For the diagnosis of OA, the KL classification is the most widely used radiographic clinical tool [36, 37]. AP knee radiographs were graded from 0 to 4, according to KL-criteria, by two independent orthopedic trauma experts (Á.C., A.M.). A grade >1 entailed an allocation to the OA group, independently from the etiology of the disease (both primary and posttraumatic cases were included). The control group consisted of patients without RA in their patient history and with a KL grade ≤1. Patients within 6 weeks of knee injury were also excluded. Pediatric patients and patients with multiple injuries, septic conditions, inadequate compliance, or incomplete dataset were also excluded.”

Patient enrollment with exact numbers is provided in the Results section (on page 5, under Patient characteristics subheading, lines 112-117):

„A total of 814 patients underwent knee joint surgery between 01 September 2019 and 31 December 2021 at the Traumatology Department of the University of Szeged. Pediatric patients (n=97), patients with infective/septic conditions (n=58), multiple injuries (n=96), compliance problems (n=29), incomplete dataset (eg. an adequate anamnesis could not be obtained) (n=24), medication with mitochondrial toxicity (eg. valproic acid) and/or potential distorting effect on our results (n=80) were excluded. In 242 cases, surgery was performed within 6 weeks post-injury, which entailed exclusion. Seventy-nine people refused to participate. Consequently, iInclusion criteria were met in 109 cases, in which 24 patients suffered from RA and 47 from OA. The control group consisted of 38 patients, without a history of OA or RA and with a need for surgery due to trauma.”

10.) Most order numbers and companies are missing.

Thank you for this comment. Missing order numbers and companies are added to the text.

 Finally, we would like to thank the Editor and the Reviewers for their conscientious work and overall positive criticism of our manuscript. We hope that the revised paper is now acceptable for publication in the Journal.